# Diversity Enhanced Narrative Question Generation for StoryBooks

**Hokeun Yoon**
Sungkyunkwan University
Suwon, South Korea
hkyoon95@g.skku.edu

**JinYeong Bak**
Sungkyunkwan University
Suwon, South Korea
jy.bak@skku.edu

## Abstract

Question generation (QG) from a given context can enhance comprehension, engagement, assessment, and overall efficacy in learning or conversational environments. Despite recent advancements in QG, the challenge of enhancing or measuring the diversity of generated questions often remains unaddressed. In this paper, we introduce a multi-question generation model (mQG), which is capable of generating multiple, diverse, and answerable questions by focusing on context and questions. To validate the answerability of the generated questions, we employ a SQuAD2.0 fine-tuned question answering model, classifying the questions as answerable or not. We train and evaluate mQG on the FairytaleQA dataset, a well-structured QA dataset based on storybooks, with narrative questions. We further apply a zero-shot adaptation on the TellMeWhy and SQuAD1.1 datasets. mQG shows promising results across various evaluation metrics, among strong baselines.[1]

## 1 Introduction

Question generation (QG), focusing on the questions derived from specific text passages or documents, plays an integral role in a wide array of domains. It improves question answering (QA) systems (Sultan et al., 2020), enriches educational experiences (Yao et al., 2022), and enhances the engagement factor in chatbots (Laban et al., 2020). The effectiveness of QG tasks can be significantly improved by generating multiple questions, ensuring a broader, more comprehensive exploration of the content.

The importance of generating and evaluating multiple questions becomes evident when we examine the creation process of QA datasets (Richardson et al., 2013; Rajpurkar et al., 2016; Xu et al., 2022). Traditional QA dataset creation typically involves

instructing annotators to create a pre-determined number of questions for a given context. Recent QG research (Wang et al., 2020a; Yao et al., 2022), however, tends to rely on automatic evaluation of semantic similarity with golden questions, often overlooking the potential for diverse aspects of questions. When generating multiple questions, diversity is a crucial aspect to consider. The diversity of questions can span several dimensions, including varied aspects of the context, different answer types, and different phrasings for essentially the same question (Karttunen, 1977). This diversity allows for a more comprehensive exploration of the context. The diversity of questions can be broadly measured based on the type of answers they require; explicit questions with answers that can be explicitly found in the reading materials, and implicit questions with answers that require deductive reasoning. The crafting of multiple questions, bearing in mind both diversity and alignment with reading materials, poses a cognitively demanding and time-consuming task for humans.

One significant application of generating diverse and multiple questions is education. It has been observed that children can develop better reading comprehension skills at an early age by creating narrative questions themselves and being asked comprehension-related questions about storybooks (Francis et al., 2005; Janssen et al., 2009). Reading comprehension is an essential skill that requires learners to combine knowledge and reason about relations, entities, and events across a given context (Kim, 2017; Mohseni Takaloo and Ahmadi, 2017). Consequently, a system that can generate diverse and multiple narrative questions can serve as a valuable enhancement to educational resources, aiding in student engagement and promoting a deeper understanding of study materials.

Recently, some researchers have attempted to generate multiple narrative questions. For educational applications, Yao et al. (2022) proposed to

---

[1]Code: https://github.com/hkyoon95/mQG

generate question-answer pairs with a three-step pipeline. As they use heuristic-generated answers to generate narrative questions most of their outcome is restricted to explicit questions. Also, Zhao et al. (2022) proposed to generate certain types of narrative questions and they tried to restrict the number of generated questions to a number of ground-truth questions, insisting that knowing question type distribution for each context is a sub-skill in education (Paris and Paris, 2003). We set these two approaches as our main baselines.

To address the above challenges, we introduce a multi-question generation model (mQG) that generates diverse and contextually relevant questions by referencing questions from the same context. mQG is trained with maximum question similarity loss $L_{MQS}$, which is designed to make the representation of reference questions and the representation of a target question similar. Moreover, mQG employs a recursive generation framework, where previously generated questions are recursively fed back into the model as mQG is trained to output different questions from reference questions. Same as our two baselines, mQG is trained and evaluated on the FairytaleQA dataset, which focuses on narrative comprehension of storybooks. This dataset is designed to provide high-quality narrative QA pairs for students from kindergarten to eighth grade (ages 4 to 14), and labeled questions as explicit or implicit. We adopt Self-BLEU (Zhu et al., 2018) to evaluate the diversity of generated questions. Beyond diversity, to consider generated questions relevant to the context, we demonstrate the answerability evaluation model to assess whether the generated questions are answerable. We also evaluate on TellMeWhy (Lal et al., 2021) and SQuAD1.1 (Rajpurkar et al., 2016) datasets with zero-shot adaptation to further analyze the performance of mQG in different settings. Differing from previous approaches, mQG successfully generates a substantial number of diverse and answerable narrative questions.

The main contributions of this paper are summarized as follows.

- We expand the scope of the question generation task by generating a comprehensive set of questions, regardless of our knowledge of the answers, and subsequently categorize them into answerable and non-answerable questions.

- We introduce mQG, a novel question genera-

tion model that is trained using the maximum question similarity loss $L_{MQS}$ and employs a recursive referencing process for generating a wide array of questions while preserving semantic correctness.

- We introduce an answerability evaluation model capable of classifying questions as implicit, explicit, or unanswerable.

## 2 Related Work

### 2.1 Question Generation

Based on given contents, question generation aims to generate natural language questions, where the generated questions are able to be addressed with the given contents. After neural approaches took over a large proportion in QG (Yuan et al., 2017; Zhou et al., 2017), QG can largely be separated by target answer aspect into answer-aware QG and answer-unaware QG. Answer-aware QG, as its name implies, provides an answer to a model and prompts it to generate questions based on those answers. On the other hand, answer-unaware QG mainly focuses on the context to formulate questions. The introduction of pre-trained Language Models (LMs) further accelerated advancements in QG, and many works have demonstrated significant improvement in the answer-aware QG task and presented promising possibilities for QG (Zhang and Bansal, 2019; Dong et al., 2019; Yan et al., 2020). This approach inherently favors explicit questions, which can be directly answered with the provided context. In answer-unaware QG, only a handful of studies have been conducted, primarily focusing on strategies such as sentence selection from a paragraph (Du and Cardie, 2017), employing transformer architectures with out-of-vocabulary methods (Scialom et al., 2019), and generating questions based on silver summaries (Zhao et al., 2022). In this paper, we utilize answer-unaware question generation, giving consideration to both the diversity and quality of explicit and implicit questions.

### 2.2 Diversity

In natural language generation (NLG), generating outputs that are not only correct but also diverse is essential. In the decoding aspect, diversity has been researched in areas such as top-k sampling (Fan et al., 2018), and nucleus sampling (Holtzman et al., 2020). These decoding methods tried to sample tokens from less likely vocabularies. Certain studies have focused on training models to

**FairytaleQA Dataset**

**Story Section (Context)**
the wild people who dwell in the south-west are masters of many black arts. they often lure men of the middle kingdom to their country by promising them their daughters in marriage, but their promises are not to be trusted. once there was the son of a poor family, who agreed to labor for three years for one of the wild men in order to become his son-in-law.

**Ground-truth Question**
Who agreed to labor for three years for one of the wild men in order to become his son-in-law?

**Ground-truth Question**
Who dwelled in the south-west and were masters of many black arts?

**Ground-truth Question**
What did the wild people do to lure men of the Middle Kingdom to their country?

**Training Process**

$$\mathcal{L} = \mathcal{L}_{MQS} + \mathcal{L}_{CE}$$

Figure 1: Overview of the training process of mQG. $Question(1)$ to $Question(m)$ refer to ground-truth questions from the same context (orange), without a ground-truth question (purple) input to BART Decoder. $QT$ and $[h]$ denote the wh-word corresponding to the target question and overall encoder representation.

yield more diverse outputs (Welleck et al., 2020; Yao et al., 2022), and on leveraging the combination of contrastive training and generation (Su et al., 2022). Recently, Sultan et al. (2020) evaluated the importance of diversity in QG, insisting that diverse and accurate questions yield better QA results. Additionally, some researchers explored diversity in QG based on relevant topic (Hu et al., 2018), content selectors with question type modeling (Wang et al., 2020b), control of question type (Cao and Wang, 2021), and difficulty level (Cheng et al., 2021). While these studies have addressed various aspects of diversity in QG, there is still considerable room for further research in this area. In this paper, we consider diversity a significant challenge in the question generation task and propose a model that can generate a wide range of answerable questions.

## 3 Method

In this section, we formalize the multi-question generation task and introduce our mQG. We first formulate our task and then explain how our model's training process incorporates a maximum question similarity loss $\mathcal{L}_{MQS}$. Finally, we provide a detailed outline of our recursive generation framework.

### 3.1 Task Formulation

The QG task in this paper aims to generate each question using a given context, question type, and the history of questions generated from the same context with the same question type. We use seven wh-words (what, when, where, which, who, why, how) as question types. Mathematically, given the context $C$, question type $QT$, and history of generated questions $H_i = (GQ_1, GQ_2, ..., GQ_{i-1})$, this task can be defined as generating a question, $\hat{GQ}$, where:

$$\hat{GQ} = \text{argmax}_{GQ_i}(Prob(GQ_i|QT, C, H_i)) \quad (1)$$

For the training process, we extract wh-words from each question by applying part-of-speech tagging with the Spacy[2] English Model. Due to the absence of a history of generated questions and an insufficient number of questions per context per question type in the FairytaleQA dataset, we utilize ground-truth questions that only share the context as the history of questions within the training process.

### 3.2 Diversity Enhanced Training

mQG is built upon BART (Lewis et al., 2020), which has demonstrated remarkable performance in various natural language processing tasks. The primary pre-training objective of BART is to reconstruct the masked input based on unmasked input. To further leverage the capabilities of the pre-trained BART, we introduce a maximum question similarity loss $\mathcal{L}_{MQS}$. This loss is designed to promote similar representations for different questions from the encoder and decoder.

As shown in Figure 1, the encoder takes in three inputs: the question type, which signifies the type of question to be generated; the context, which pro-

---

[2]https://spacy.io/

vides the necessary information for question generation; and ground-truth questions from the same context, serving as reference questions. These three inputs are concatenated, with a [SEP] token inserted between them. The encoder processes the input sequence and produces its corresponding representations. Subsequently, the decoder generates the representation for the target question. To calculate the maximum question similarity loss $\mathcal{L}_{MQS}$, we use mean pooling layers to convert question representations into sentence-level representations. The maximum question similarity loss $\mathcal{L}_{MQS}$ is calculated between the sentence-level representation of the reference questions and the sentence-level representation of a generated question. By encouraging the representation of different questions to be similar, we promote the generation of diverse questions that differ from reference questions.

Given a set of reference questions sentence-level representation as $Q = \{Q_1, ..., Q_m\}$ and a sentence-level representation of the target question as $TQ$, the maximum question similarity loss $\mathcal{L}_{MQS}$ is computed as follows:

$$\mathcal{L}_{MQS} = \frac{1}{m} \sum_{i=1}^{m} \max(0, 1 - s(Q_i, TQ)) \quad (2)$$

where $s(Q_i, TQ)$ is a cosine similarity calculation between representations. By optimizing the model parameters to maximize the sentence-level similarity between these different representations, we guide mQG to generate diverse questions within the range of semantic correctness. This is achieved by ensuring that all the representations, which are the ground truth questions, are semantically correct. In doing so, we maintain a balance between diversity and accuracy in the generated questions. The overall training objective $\mathcal{L}$ is defined as

$$\mathcal{L} = \mathcal{L}_{CE} + \mathcal{L}_{MQS} \quad (3)$$

$\mathcal{L}_{CE}$ refers to the cross-entropy loss from a target question. As cross-entropy loss is calculated at the token level, the use of cross-entropy loss enhances mQG to generate syntactically correct questions.

### 3.3 Recursive Generation Framework

Figure 2 illustrates the generation process of mQG. First, the encoder takes question type, and context as input. The decoder then generates a question based on the information provided by the encoder.

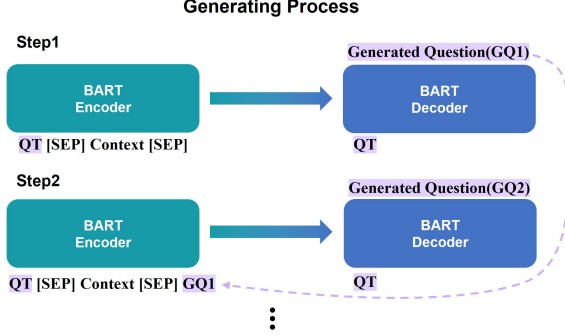

Figure 2: The Recursive Generation Framework of mQG. This framework involves an iterative process, using previously generated questions as input for subsequent steps, thereby creating a recursive cycle. Each iteration maintains the use of the same question type.

For the subsequent generation steps, the previously generated questions are recursively fed back into the model. Specifically, the previous questions are concatenated with the same question type and context, separated by a [SEP] token. This concatenated sequence is then used as input for the next generation step. This recursive generation process continues until the desired number of questions per context per question type is achieved.

The use of this recursive generation process allows mQG to generate multiple questions while considering the previously generated questions. Following the training process of mQG, this generation process enables mQG to build upon its own previous outputs and generate different questions from previous outputs. We use beam search for the decoding method and return multiple sequences to exclude pre-generated questions. By leveraging a recursive framework, mQG demonstrates its capability to generate a variety of diverse questions that are contextually relevant and coherent.

## 4 Experiments

### 4.1 Dataset

**FairytaleQA** (Xu et al., 2022). We train mQG with the FairytaleQA dataset, which is constructed for educational purposes. Each book is split into sections and annotators were instructed to create on average 2-3 narrative question-answer pairs per section. All question-answer pairs are annotated based on seven question types that capture narrative elements/relations. Questions are labeled as explicit or implicit questions based on whether or not the answer source can be directly found

in the context. The original FairytaleQA dataset is constructed in a train/validation/test set with 232/23/23 books and 8,548/1,025/1,007 QA pairs. From the entire dataset, a small portion of questions (985 out of 10,580) spans multiple paragraphs. As mQG and baselines are fit for one paragraph we remove those questions. To cross-validate, we randomly shuffled the dataset and split it by books in train/validation/test set with roughly matching 80/10/10 (%).

## 4.2 Baselines

In the experiments, we compare mQG with four baselines; an end-to-end model initialized with BART-large, and methods proposed in Su et al. (2022), Yao et al. (2022), Zhao et al. (2022) denoted as CB, QAG, and EQG. The last two baselines are designed for multiple question generation purposes.

**E2E.** As the FairytaleQA dataset consists of multiple questions in one context, we concat all questions and train the BART-large model to generate questions based on each context. To match the number of generated questions, we set the maximal target length to 280 tokens which roughly matches the number of generated questions setting of mQG.

**CB (Contrastive Baseline).** We construct this baseline following the framework in Su et al. (2022), which tackles the problem of diversity in open-ended text generation. This framework first trains the language model using contrastive loss and decodes it with a contrastive search method. Since the contrastive baseline is proven for diverse text generation we apply it to GPT2 (denoted as CB (GPT2)), and BART (denoted as CB (BART)) and set it as our baseline. During generation, the maximal target length is set to 280 tokens.

**QAG.** This baseline follows a question-answer generation architecture by Yao et al. (2022). This architecture first generates answers based on a heuristic-based answer generation module, which generates multiple answers per context. With the generated answers, BART generates corresponding questions. And, to verify the quality of the generated questions, DistilBERT ranking module ranks each QA pair and chooses the top questions. As our task is to generate multiple questions, we

denote architecture without a ranking module as QAG and the top 10 questions per context chosen by the ranking module as QAG (top 10).

**EQG.** EQG model (Zhao et al., 2022) generates questions based on silver summaries. Silver summary is a method proposed by Demszky et al. (2018), which inserts answers into the semantic parsed questions with a rule-based method. EQG consists of three steps: 1) generate question type distribution for each context with BERT; 2) generate silver summary with BART, using question type, question type ordering from a question type distribution module, and context; 3) generate question based on silver summary, question type, and question ordering with BART. Without a question type distribution module, EQG is able to generate multiple questions. Since our approach is to generate multiple questions we set the EQG baseline without question type distribution module.

## 4.3 Automatic Evaluation

### 4.3.1 Evaluation Metrics

In evaluating question generation, both the quality and diversity of the generated questions are critical components. Thus, we evaluate each aspect with separate automatic evaluation metrics. We use Rouge-L score (Lin, 2004), BERTScore (Zhang et al., 2020), and BLEURT (Sellam et al., 2020) to measure the quality of generated questions. Similar to Yao et al. (2022), for each ground-truth question, we find the highest semantic similarity score on generated questions from the same context than average overall semantic similarity scores. And, with multiple questions generated from the same context, we recognize the necessity to measure diversity automatically. For diversity measurement, we use Self-BLEU score (Zhu et al., 2018) which was introduced to evaluate just a variety of sentences. The Self-BLEU score, which uses each generated sentence as a hypothesis and others as references, is employed to evaluate the diversity of questions generated from the same context. A lower Self-BLEU score represents greater diversity. All metrics ranges are between 0 to 1 except Rouge-L score (0 to 100).

### 4.3.2 Answerability Evaluation Model

In order to evaluate whether the generated questions correspond to the context, we leverage SQuAD2.0 dataset (Rajpurkar et al., 2018) to build

| | FairytaleQA | | | | | | | | | | | | |
|---|---|---|---|---|---|---|---|---|---|---|---|---|---|
| **Architecture** | **# Generated Questions Per Section** | | **# Answerable Questions Per Section ↑** | | **Rouge-L F1 ↑** | | **BERTScore F1 ↑** | | **BLEURT ↑** | | **Self-BLEU ↓** | |
| | M | SE | M | SE | M | SE | M | SE | M | SE | M | SE |
| E2E | 1.58 | 0.07 | 1.45 | 0.12 | 36.05 | 0.35 | 0.8960 | 0.0062 | 0.4064 | 0.0104 | - | - |
| CB (BART) | 1.60 | 0.03 | 1.49 | 0.04 | 36.89 | 0.68 | 0.9074 | 0.0017 | 0.4045 | 0.0072 | - | - |
| CB (GPT2) | 3.28 | 0.43 | 0.96 | 0.56 | 26.47 | 1.27 | 0.8937 | 0.0020 | 0.3328 | 0.0077 | 0.8906 | 0.0120 |
| EQG | 28.00 | 0.00 | 3.80 | 0.76 | 41.05 | 1.61 | 0.9136 | 0.0034 | 0.4293 | 0.0118 | 0.9864 | 0.0043 |
| QAG (top10) | 9.95 | 0.14 | 6.57 | 0.39 | 45.44 | 0.81 | 0.9208 | 0.0006 | 0.4444 | 0.0076 | 0.7608 | 0.0078 |
| QAG | 26.97 | 0.50 | 15.95 | 1.24 | 53.77 | 1.03 | 0.9323 | 0.0009 | 0.5140 | 0.0115 | 0.8874 | 0.0030 |
| mQG | 28.00 | 0.00 | **23.08** | 0.36 | **58.90** | 0.37 | **0.9394** | 0.0005 | **0.5698** | 0.0033 | **0.6389** | 0.0079 |

Table 1: Three cross-validation results on the FairytaleQA dataset. # Answerable Questions Per Section is based on the answerability evaluation model, as described in section 4.3.1. ↑ means higher is better, ↓ means lower is better. Due to a low number of questions, Self-BLEU which cannot be measured is marked with a hyphen. M, SE denotes mean and standard error. mQG generates the highest number of answerable questions with greater diversity.

an evaluation model. SQuAD2.0 is a question-answering dataset with 100K answerable questions and 50K unanswerable questions. This dataset is used to enhance the evaluation model by classifying whether the questions are answerable or not. We use DeBERTa-base (He et al., 2021) as the backbone model.

To achieve our goal, we train the evaluation model on the QA task following implementation in Devlin et al. (2019). We construct two dense layers above the encoder; one for the answer start position and the other for the answer end position. And, as unanswerable questions and implicit questions do not have an answer span, for these questions [CLS] token is assigned as the answer start position and the answer end position. For implicit questions in the FairytaleQA dataset, we add a special token [IMP] and assign it as an answer start span and answer end span. First, we train the evaluation model with the SQuAD2.0 dataset on the QA task. For the second step, we train the evaluation model again with the FairytaleQA dataset. By utilizing a two-step training, the evaluation model is able to classify generated questions as explicit, implicit, or unanswerable. The number of answerable questions per section in Table 1 are based on classified results by the evaluation model. If the evaluation model classifies generated questions as implicit or explicit, then we count them as answerable. (Answerability evaluation model details are given in Appendix A.)

### 4.3.3 Results

Table 1 presents evaluation results on the Fairy-taleQA test set. '# Generated Questions Per Section' refers to the number of questions generated

for each section. In '# Answerable Questions Per Section', as duplicate questions within the same context are not needed, we leave only one question from duplicate questions. Even though mQG is able to generate multiple questions within the maximum token length of BART, we roughly match the number of questions to QAG for fair comparison in Rouge-L F1, setting mQG to generate 4 questions per section per question type, totaling 28 questions per section. The same setting is applied to EQG, as EQG does not have limitations in generating multiple questions.

General baselines (E2E and CB) that generate multiple questions in one iteration show significant underperformance in the Rouge-L F1 score and in the number of generated questions, compared to strong baselines (QAG and EQG), and the mQG. This indicates that to generate multiple questions, a specific model is needed. Across all evaluation metrics, mQG consistently outperforms the baselines.

### 4.4 Human Evaluation

We evaluate the diversity and quality of generated questions on the FairytaleQA dataset with human judges. We hire five annotators, proficient in English as their first foreign language, to further evaluate the diversity and quality of the generated questions. We follow the human evaluation procedure described by Cao and Wang (2021) and compare mQG, with two robust baselines, EQG and QAG.

**Question Diversity.** In the question diversity study, we randomly sample 5 books from the

| Architecture | Type (%) | Syntax (%) | Content (%) |
|---|---|---|---|
| EQG | 22.0 | 18.0 | 23.5 |
| QAG | 33.0 | 22.0 | 34.5 |
| mQG | **77.0** | **70.5** | **60.0** |

Table 2: Human evaluation on diversity. The percentage of samples ranked first among other models. Krippendorf's alphas are 0.69, 0.51, and 0.38 for the three dimensions. Ties are allowed. mQG demonstrates the most diversity in all dimensions.

| Architecture | Appro. | Ans. |
|---|---|---|
| EQG | **4.85** | 4.46 |
| QAG | 4.60 | 4.43 |
| mQG | 4.79 | 4.47 |
| Ground-truth | 4.71 | **4.76** |

Table 3: Human evaluation on appropriateness (Appro.) and answerability (Ans.). The Krippendorf's alphas are 0.14 and 0.27 for the two dimensions. Ties are allowed. In all models, not much difference is observed compared to ground truth questions.

original test set; and for each book, we randomly sample 8 sections, totaling 40 sections. For each section, we randomly sample three questions as a question set from each model, and provide only the question sets for annotation. For each question set, the annotators rank the three models on a scale of 1 (highest) to 3 (lowest) based on three dimensions of diversity: type–whether the three selected questions have different question types; syntax–whether the three selected questions use different syntax; and content–whether the three selected questions need to be addressed with diverse answers.

As shown in Table 2, on all dimensions, human annotators rate mQG as generating the most diverse questions compared to the other models, with each question requiring a different answer.

**Question Quality.** In the question quality study, we again randomly sample 5 books from the original test set. For each book, we select a random sample of 8 sections. Each section contains four questions, each randomly sampled from three models and ground-truth, totaling 160 questions. Two dimensions are rated from 1 (worst) to 5 (best): appropriateness–whether the question is semantically correct; answerability–whether the question can be addressed by a given section.

As shown in Table 3, all models, when compared to the ground-truth, generate semantically correct questions. Given that mQG can generate a broad diversity of questions, these results confirm that mQG fulfills our goal of generating multiple questions while maintaining semantic correctness and relevance to the context.

### 4.5 Zero-shot Performance Evaluation

We conduct a zero-shot evaluation on two distinct datasets, to test mQG more in various real-world scenarios, where contexts and desired questions can differ. Zero-shot evaluation is essential for assessing model performance as it illuminates the model's ability to generalize beyond the specific examples it was trained on.

#### 4.5.1 Dataset

**TellMeWhy (Lal et al., 2021).** TellMeWhy dataset comprises free-form why-questions related to events in short sections. The dataset was created using template-based transformations to generate questions, with crowdsourcing to gather answers. Sections were sourced from ROCStories (Mostafazadeh et al., 2016), a similar domain to the training dataset (FairytaleQA). TellMeWhy contains a mixture of explicit and implicit questions. Approximately 28.82% of questions in the dataset are implicit. We evaluate with 1,134 sections and 10,689 questions from the test split.

**SQuAD1.1 (Rajpurkar et al., 2016).** Squad1.1 dataset is a comprehensive benchmark that focuses on machine comprehension, question generation, and question answering tasks. It consists of a large collection of articles from Wikipedia, covering a wide range of topics, which is a different source from the training dataset (FairytaleQA). Each article is accompanied by a set of only explicit questions. We evaluate with 2,429 sections, and 12,010 questions from the SQuAD1.1 test split created by Du et al. (2017).

#### 4.5.2 Zero-shot Results

In zero-shot evaluation, we compare mQG with two strong baselines, EQG and QAG. Initially, we examine the performance on the Tellmewhy dataset in Table 4. Given that the TellMeWhy dataset only contains why-questions, we select why-questions from the generated questions for evaluation. mQG achieved the highest semantic similarity scores and outperformed baseline models in terms of the number of answerable questions and exhibited better

**TellMeWhy**

| Architecture | # Generated Questions Per Section | # Answerable Questions Per Section ↑ | Rouge-L F1 ↑ | BERTScore F1 ↑ | BLEURT ↑ | Self-BLEU ↓ |
|---|---|---|---|---|---|---|
| EQG | 4.00 | 0.63 | 35.91 | 0.9129 | 0.4126 | 0.9425 |
| QAG | 1.53 | 0.45 | 30.35 | 0.9231 | 0.4360 | - |
| mQG | 4.00 | **2.10** | **56.17** | **0.9361** | **0.5475** | **0.3191** |

Table 4: Zero-shot evaluation result on TellMeWhy dataset. Due to a low number of questions, Self-BLEU which cannot be measured is marked with a hyphen. mQG shows the highest semantic similarity scores with more diversity and generates the largest number of answerable questions.

**SQuAD1.1**

| Architecture | # Generated Questions Per Section | # Answerable Questions Per Section ↑ | Rouge-L F1 ↑ | BERTScore F1 ↑ | BLEURT ↑ | Self-BLEU ↓ |
|---|---|---|---|---|---|---|
| EQG | 28.00 | 3.74 | 30.31 | 0.8977 | 0.4219 | 0.9695 |
| QAG | 29.77 | 14.40 | **46.75** | 0.9203 | 0.5265 | 0.7172 |
| mQG | 28.00 | **20.15** | 45.38 | **0.9211** | **0.5508** | **0.6157** |

Table 5: Zero-shot evaluation result on SQuAD1.1 dataset. mQG generates the most answerable questions with more diversity.

diversity. Zero-shot evaluation on the Tellmewhy dataset, which contains a mix of explicit and implicit questions, demonstrates the ability of mQG to generate different question styles based on answers effectively.

Table 5 shows evaluation results on the SQuAD1.1 dataset. Even with an out-of-domain dataset, mQG still demonstrates notable performance. mQG outperforms in generating diverse questions and producing a greater number of answerable questions compared to other baselines. However, in the Rouge-L F1 score, mQG is slightly lower than QAG. This can be attributed to the exclusive focus of the SQuAD dataset on explicit questions, and the answer-aware question generation method used by QAG, which is renowned for its effectiveness in generating explicit questions. Yet, when employing embedding-based evaluation methods such as BERTScore and BLEURT, mQG outperforms the baseline models, particularly in the case of BLEURT. The fact that mQG still demonstrates decent performance on the SQuAD dataset, despite the limitation of the dataset to explicit questions and its status as an out-of-domain dataset, further emphasizes the effectiveness of mQG.

Through these two different settings, we see promising results of mQG. It shows the adaptability of mQG to diverse question styles and domains, further validating the robustness and utility of mQG.

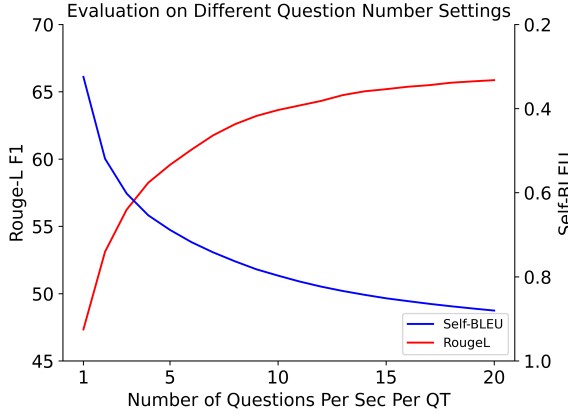

Figure 3: Results of different question number settings on the original FairytaleQA test set. Self-BLEU is presented here in a reversed format to allow for a more intuitive visual comparison. Intersections of the curves represent the optimal trade-off between two metrics.

## 5 Ablation Study

### 5.1 Setting of Question Number

Given that mQG can be set with the number of questions to generate, we conduct an experiment on various settings of question number per section per question type to generate. In Figure 3, the evaluation result is based on the original FairytaleQA test set. As the quantity of generated questions increases, the Rouge-L F1 score provides satisfactory results, though diversity decreases. This indicates

| | FairytaleQA | | | | | | | | | | | |
|---|---|---|---|---|---|---|---|---|---|---|---|---|
| Architecture | # Generated Questions Per Section | | # Answerable Questions Per Section ↑ | | Rouge-L F1 ↑ | | BERTScore F1 ↑ | | BLEURT ↑ | | Self-BLEU ↓ | |
| | M | SE | M | SE | M | SE | M | SE | M | SE | M | SE |
| mQG | 28.00 | 0.00 | **23.08** | 0.36 | **58.90** | 0.37 | **0.9394** | 0.0005 | 0.5698 | 0.0033 | **0.6389** | 0.0079 |
| - $\mathcal{L}_{MQS}$ | 28.00 | 0.00 | 22.67 | 0.28 | 58.66 | 0.08 | **0.9394** | 0.0003 | **0.5703** | 0.0019 | 0.7006 | 0.0045 |
| - $\mathcal{L}_{MQS}$ & reference questions | 28.00 | 0.00 | 22.65 | 0.41 | 54.76 | 0.22 | 0.9353 | 0.0005 | 0.5428 | 0.0011 | 0.7529 | 0.0032 |

Table 6: The comparison results of mQG with and without maximum question similarity loss and reference questions.

that a significant increase in the number of generated questions tends to produce similar questions with different phrasings. Setting the number of generated questions at 4 shows the optimal trade-off between the Rouge-L F1 and the Self-BLEU.

## 5.2 Analysis of Maximum Question Similarity Loss and Recursive Framework

As discussed in section 5.2, mQG aims to increase diversity within questions while maintaining semantic correctness. mQG w/o $\mathcal{L}_{MQS}$ refers to the mQG model only trained with $\mathcal{L}_{CE}$. For mQG w/o $\mathcal{L}_{MQS}$ and reference questions, we give only question type and context as input while training, and no recursive framework is used in inference. Table 6 shows that the mQG model with maximum question similarity loss $\mathcal{L}_{MQS}$ and reference questions hugely increase diversity. Additionally, the number of answerable questions has also improved. This could be attributed to the fact that all ground-truth questions are answerable, and mQG maximizes the similarity between these questions and continually references the most probable question during inference. These results indicate that each framework of mQG effectively enhances the probability of generating a diverse set of possible questions.

## 6 Conclusion

In this work, we extend the scope of answer-unaware question generation to generate multiple diverse questions. We propose a novel framework that applies a maximum question similarity loss during training to promote question diversity, followed by a recursive generation process for further refinement. Additionally, an evaluation model is introduced to verify the answerability of the generated questions. Recognizing the essential role of narrative questions in education, we train and evaluate mQG accordingly. Comprehensive experiments validate the efficacy of mQG across a variety of datasets, highlighting its potential utility in environments that demand diverse narrative questions.

## Limitations

mQG framework utilizes a recursive feedback mechanism for generating questions during the inference stage. However, the quality of these generated questions remains uncertain. If the quality of previously generated questions is poor, this may adversely impact the quality of subsequent questions produced by mQG. Moreover, the quantity of questions that can be generated is limited by a maximum token threshold. Another limitation is the potential risk of misclassification by the evaluation model, which could lead to the categorization of unanswerable questions as answerable. Despite our efforts to mitigate this risk, the evaluation model is still at a level of uncertainty in accurately classifying the generated questions. Even with the fact that reliability scores can be low in NLP tasks, in the quality human evaluation, the reliability scores are relatively low. This can lead to uncertainty in the results.

## Ethics Statement

The results are appropriately placed in the context of prior and existing research. All generation models are trained on the FairytaleQA dataset which is publicly available and has no ethical issues as annotated by educational experts. In the human evaluation process, we pay annotators more than the minimum wage.

## Acknowledgements

We would like to thank the anonymous reviewers for their helpful questions and comments. JinYeong Bak is the corresponding author. This work was partly supported by Institute of Information & communications Technology Planning & Evaluation (IITP) grant funded by the Korea

government (MSIT) (No.2022-0-00680, Abductive inference framework using omni-data for understanding complex causal relations & No.2019-0-00421, AI Graduate School Support Program (Sungkyunkwan University)), and a grant from the National Research Foundation of Korea (NRF) [NRF-2021R1A4A3033128].

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

# Appendix

# A    Further Analysis on Evaluation Model

## A.1    Preprocessing Dataset

To evaluate each cross-validation set with an answerability evaluation model, we train the evaluation model with different FairytaleQA trainsets. One is an originally constructed trainset and the others are randomly split by books. From the FairytaleQA dataset, some explicit questions were not able to be found in the section and some questions

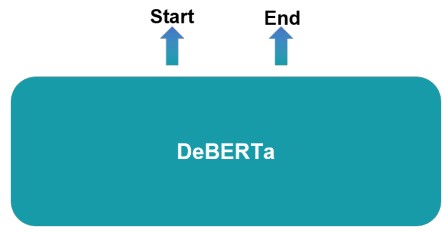

[CLS] **[IMPLICIT]** question [SEP] context

Figure 4: Overview of Answerability Evaluation Model.

|             | Explicit | Implicit | Total |
|-------------|----------|----------|-------|
| # questions | 5,376    | 1,963    | 7,309 |

Table 7: The number of questions of the FairytaleQA dataset after annotation mistakes were removed.

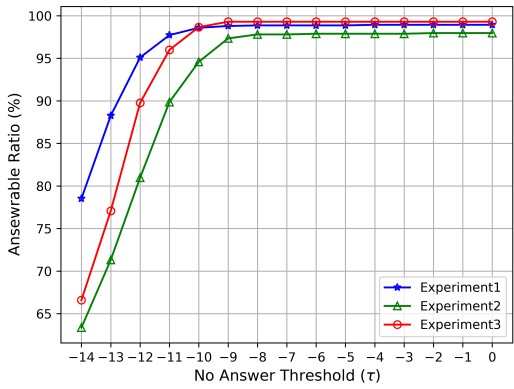

Figure 5: The answerable ratio of val+test set by different threshold settings.

with cross-annotated answers had different aspects of answers (explicit, implicit). We removed those questions and a number of total questions after pre-processing is described in Table 7.

## A.2    Evaluation Model Postprocessing

In terms of post-processing, we take a similar approach by Devlin et al. (2019). Classified results $y_c$ of each question are formulated as:

$$
y_c = \begin{cases} \text{No Answer,} & \text{if } CLS_{se} > a_{se} + \tau \\ & \text{and } CLS_{se} > IMP_{se} + \tau \\ \text{Implicit,} & \text{else if } IMP_{se} > a_{se} \\ \text{Explicit,} & \text{otherwise.} \end{cases}
$$
(4)

$CLS_{se}$ denotes score of [CLS] token as answer start span and answer end span. $IMP_{se}$ denotes score of [IMP] token as answer start span and answer end span. $a_{se}$ denotes the best score of answer

| | F1 | | Accuracy | |
|---|---|---|---|---|
| | M | SE | M | SE |
| Explicit | 78.72 | 0.52 | 88.26 | 0.17 |
| Implicit | 64.76 | 2.05 | 64.76 | 2.05 |
| Total | 75.28 | 1.01 | 82.49 | 0.81 |

Table 8: Ground-truth val+test set results on the evaluation model. Each model is trained with each cross-validation trainset.

| | FairytaleQA | | | | |
|---|---|---|---|---|---|
| | Ground-truth | QAG (top10) | QAG | EQG | mQG |
| Explicit | 74.10% | 79.05% | 71.69% | 54.42% | 60.65% |
| Implicit | 21.22% | 2.50% | 5.08% | 33.95% | 22.38% |
| No Ans. | 4.68% | 18.45% | 23.23% | 11.63% | 16.97% |
| Total | 919 | 2,835 | 7,534 | 1,402 | 8,820 |

Table 9: The FairyTaleQA test set analysis of questions by answer types, classified by evaluation model. Total denotes the number of questions after duplicates from the same context are removed. Each answer type is denoted with a proportion in each model.

start span and answer end span without [CLS] and [IMP]. Additionally, if an answer end span indice is lower than an answer start span indice we classify it as no answer. Threshold $\tau$ is selected on the ground-truth set to maximize the performance. This threshold is set differently for each evaluation model. Figure 5 shows the answerable ratio percentage by different threshold settings. We also train three evaluation models with each train set for cross-validation in the main results. We select each threshold before a significant drop in the answerable ratio is observed. -12, -10, and -11 are each threshold for experiment1, experiment2, and experiment3.

## A.3 Evaluation Model Results

We perform cross-validation to measure the performance of the main results in Table 1, and as a result, we train each evaluation model with each trainset. Since our goal is to classify questions as explicit, implicit, or unanswerable, we count explicit questions as accurate if at least one of the predicted answer tokens is found in the ground-truth answer. This is denoted as "Accuracy" in Table 8. The F1 measurement follows the implementation by Devlin et al. (2019). The evaluation model classifies explicit questions more accurately than implicit questions.

| Self-BLEU | Example Questions |
|---|---|
| 0.3150 | Why did the Dragon King want to capture a monkey? Why couldn't the Dragon King's servants capture a monkey? Why did the Dragon King consult his chief steward? Why was the Dragon King greatly puzzled? |
| 0.6362 | Why did the Dragon King want to capture a monkey? Why couldn't the Dragon King's servants capture a monkey? Why did the Dragon King consult his chief steward? How did the Dragon King consult his chief steward? |
| 0.7830 | Why did the Dragon King consult his chief? Why did the Dragon King consult steward? Why did the Dragon King consult his chief steward? How did the King consult his chief steward? |
| 0.9014 | Why did the Dragon King consult his chief steward? Why did the Dragon King consult his chief? Why did the Dragon King consult his chief steward? How did the Dragon King consult his chief steward? |

Table 10: Examples on Self-BLEU scores with 4 questions each.

## A.4 Classified Questions Analysis

We analyze the ratio of questions classified into different answer types by the answerability evaluation model. Even though the ground-truth questions do not contain unanswerable questions, the evaluation model classifies approximately 4.5% of the questions as unanswerable, as shown in Table 5. The problem of answer-aware question generation is well-known. QAG uses the answer as an input in the question generation process, and our results show that QAG is not fit for generating implicit questions, as only about 5.1% of questions are classified as implicit. The EQG baseline generates both explicit and implicit questions but only has a small number of total questions after removing duplicates. On the other hand, the mQG still has a large number of questions even after removing duplicates, totaling 8,820, with explicit and implicit questions roughly in a 3-to-1 ratio. These results show that the mQG generates both types of multiple questions better than other baselines.

## B Diversity Exploration

For diversity evaluation, we calculate the Self-BLEU score among generated questions from the same context. Self-BLEU score is based on BLEU evaluation method (Papineni et al., 2002). The BLEU evaluation method has many criticisms for evaluating sentence-level corpus. If a higher-order n-gram precision goes to 0, the total BLEU score goes to 0. As an outcome, many variations applying the smoothing method for the BLEU score have shown (Chen and Cherry, 2014). We apply 'smoothing 1' described in Chen and Cherry (2014) since all the generated questions are sentence-level.

| FairytaleQA | | | | |
|---|---|---|---|---|
| Architecture | Decdoing Method | # Answerable Questions Per Section ↑ | Rouge-L F1 ↑ | Self-BLEU ↓ |
| mQG-T5 | b=5 | 17.89 | 30.59 | **0.5476** |
| mQG-BART | b=5 | **23.35** | **58.24** | 0.6243 |
| | p=0.1 | 16.89 | 53.45 | 0.7826 |
| | p=0.5 | 18.01 | 53.54 | 0.7622 |
| | p=0.75 | 19.12 | 54.45 | 0.7321 |
| | p=0.95 | 20.06 | 54.90 | 0.7135 |

Table 11: Performance of mQG with different backbone models and decoding methods on the original test set. b=5 denotes beam search with beam size set to 5. p denotes nucleus sampling (NS@p; $p \in 0.1, 0.5, 0.75, 0.95$). All models are set to generate 28 questions per section.

Examples of Self-BLEU scores are shown in table 10. When the Self-BLEU score goes up to 0.7830, almost all questions can be addressed by the same answers.

## C Decoding Method and Model Selection

Moreover, in addition to the main results, we compare the performance of mQG between different backbone models and decoding methods. In Table 11, T5-based mQG exhibits the best Self-BLEU score but significantly lags behind BART-based mQG in terms of # Answerable Questions Per Section and Rouge-L score. This suggests that T5-based mQG struggles to generate semantically correct questions. When comparing decoding methods, beam search outperforms nucleus sampling in all dimensions. This is due to the decoding process of mQG, which returns multiple sequences to exclude pre-generated questions. Beam search utilizes a tree search algorithm, whereas nucleus sampling does not. As a result, nucleus sampling tends to generate duplicate questions.

## D Weighting Factor Impact on Performance

To determine how MQS loss affects training, we conduct experiments with the mQG model using different settings for the weighting factor $\beta$. The overall training objective $\mathcal{L}$ is defined as

$$\mathcal{L} = \mathcal{L}_{CE} + \beta * \mathcal{L}_{MQS} \quad (5)$$

In Table 12, Self-BLEU is calculated between questions that share context and question type. The optimal point of diversity is achieved when $\beta$ is set to 0.4. As $\beta$ increases, the Self-BLEU score decreases, while the number of answerable questions increases. This outcome aligns with our goal

| FairytaleQA | | | |
|---|---|---|---|
| $\beta$ | # Answerable Questions Per Section ↑ | Rouge-L F1 ↑ | Self-BLEU ↓ |
| 0.0 | 22.89 | 58.49 | 0.4747 |
| 0.2 | 23.16 | 59.40 | 0.4117 |
| 0.4 | 23.23 | **59.54** | **0.4052** |
| 0.6 | 23.26 | 58.44 | 0.4261 |
| 0.8 | 23.34 | 59.29 | 0.4288 |
| 1.0 | 23.35 | 58.24 | 0.4210 |
| 2.0 | 23.28 | 58.28 | 0.4297 |
| 3.0 | 23.34 | 58.42 | 0.4478 |
| 5.0 | **23.50** | 58.15 | 0.4527 |

Table 12: mQG results on different $\beta$ settings on the original test set. 0.0 equals to mQG w/o maximum question similarity loss $\mathcal{L}_{MQS}$. All models are set to generate 28 questions per section.

| Architecture | Rouge-L (ori) | Rouge-L (alt) | Diff |
|---|---|---|---|
| **FairytaleQA** | | | |
| EQG | 41.05 | 39.35 | 1.70 |
| QAG | 53.77 | 53.13 | 0.64 |
| mQG | 58.90 | 58.36 | 0.54 |
| **TellMeWhy** | | | |
| EQG | 35.91 | 15.08 | 20.83 |
| QAG | 30.35 | 23.93 | 6.42 |
| mQG | 56.17 | 51.57 | 4.60 |
| **SQuAD1.1** | | | |
| EQG | 30.31 | 25.84 | 4.47 |
| QAG | 46.75 | 44.85 | 1.90 |
| mQG | 45.38 | 43.20 | 2.18 |

Table 13: Comparison results on Rouge-L calculation. FairytaleQA results are the mean value of 3 cross-validation results. Rouge-L (alt) denotes one-to-one match calculation. Diff denotes the difference between Rouge-L (ori) and Rouge-L (alt).

of implementing MQS loss to enhance diversity within the bounds of semantic correctness.

## E Another Rouge-L Calculation

As mentioned in Section 4.3, we calculate the Rouge-L score only to find the highest score for each ground-truth question. This calculation method may lead to the one-to-many matching problem. To determine if the problem has occurred, we compare the results with another Rouge-L calculation Rouge-L (alt). This calculation excludes previously matched generated questions, allowing for only one-to-one matches. In Table 13, most Rouge-L (alt) results exhibit slightly lower scores in comparison to Rouge-L (ori), suggesting that

one-to-many problems have occurred, although the impact is relatively minor as the ground-truth questions are a unique set of questions. The significant difference in the TellMeWhy dataset can be attributed to the limited number of 'why' questions generated.

## F    Implementation Details

For the mQG model, we use the MQS loss of the validation set as the selecting criteria. For the mQG models without MQS loss, we use MLE loss as the selecting criteria. Total training time was about 3 hours with 1 RTX A6000 GPU. We initialize the mQG model with pretrained BART-large, which has 406M parameters. Hyperparameters are follow: learning rate = 5e-6; batch size = 8; epoch = 15

We use RoBERTa-large model for BERTScore and BLEURT-20 model for BLEURT. For the evaluation model, we load SQuAD 2.0 finetuned DeBERTa-base model [3], which has 86M parameters, to further finetune.   Total training time was about an hour with 1 RTX A6000 GPU. Hyperparameters are follow: learning rate = 5e-6; batch size = 16; epoch = 8

## G    Examples of Generated Questions

Tables 14 and 15 show the generated examples of the mQG, EQG, QAG, and ground truth questions with the according section and classified results with the answerability evaluation model. Even with different settings for generating multiple questions, EQG still generated duplicate questions because it guided the model only with special tokens to generate multiple questions. QAG has generated different questions but with less diversity. In all questions, the evaluation model accurately classified the questions. Given the sufficient number of questions generated by each model, we selected four questions as representative examples. Given the sufficient number of questions generated by each model, we selected 4 questions as representative examples.

---

[3]https://huggingface.co/deepset/deberta-v3-base-squad2

**Section**

But his brother complained of being weary, and at length they decided to remain there for the night. When Andrew awoke he found himself alone; and he saw neither brother nor boat, until he came to the highest point of the island. Then he discovered him far out, darting for land like a sea-gull. Andrew did not understand the whole affair. There were still provisions there, as well as a dish of curd, his gun and various other things. So Andrew wasted but little time in thought. "He will come back this evening," said he. "Only a fool loses heart so long as he can eat." But in the evening there was no brother to be seen, and Andrew waited day by day, and week by week; until at last, he realized that his brother had marooned him on this barren island in order to be able to keep their inheritance for himself, and not have to divide it. And such was the case, for when John Nicholas came in sight of land on his homeward trip, he had capsized the boat, and declared that Lucky Andrew had been drowned.

**Ground-truth Questions**

What was John Nicholas doing when Andrew saw him? (Explicit)

Why did John Nicholas capsize the boat when he reached land? (Implicit)

Why did Andrew want the inheritance to himself? (Implicit)

How did Andrew feel when he saw his brother and boat far out? (Implicit)

**mQG**

What did John Nicholas declare when he came in sight of land? (Explicit)

Why did John Nicholas marooned his brother on a barren island? (Explicit)

Why did the brother want to keep their inheritance for himself? (Implicit)

Why did John Nicholas declare that Lucky Andrew had been drowned? (Implicit)

**QAG**

what did andrew find on the island? (Explicit)

what did andrew find on the island? (Explicit)

who did andrew see after he woke up? (No Answer)

when andrew awoke he found himself alone? (No Answer)

**EQG**

What happened after John Nicholas came in sight of land on his homeward trip? (Explicit)

What happened after John Nicholas came in sight of land on his homeward trip? (Explicit)

Why did the brother complain of being weary? (Implicit)

Why did the brother complain of being weary? (Implicit)

Table 14: Example of generated questions of mQG, EQG, and QAG for a given context. On the right of each question, classified results by the answerability evaluation model are shown in parentheses. Generated questions of mQG have the most diverse syntax questions considering question types and still, semantically correspond to the given section.

**Section**

At any rate, his poor young wife was sorely to be pitied, for she was left with a little baby-boy to bring up, and, as times were bad, she had not much to live on. But she loved her baby dearly, and worked all day amongst her cows, and pigs, and hens, in order to earn enough money to buy food and clothes for both herself and him. Now, on the morning of which I am speaking, she rose very early and went out to feed her pigs, for rent-day was coming on, and she intended to take one of them, a great, big, fat creature, to the market that very day, as she thought that the price that it would fetch would go a long way towards paying her rent. And because she thought so, her heart was light, and she hummed a little song to herself as she crossed the yard with her bucket on one arm and her baby-boy on the other.

**Ground-truth Questions**

What did the widowed Mistress do to earn enough money? (Explicit)

Why did the widowed Mistress want to sell her pig? (Explicit)

How did the widowed Mistress feel while she thought of selling the pig? (Implicit)

**mQG**

What did the young woman do because rent-day was coming on? (Explicit)

Why did the young wife want to buy a great, big, fat creature? (Explicit)

Where did the young woman intend to take one of the pigs? (Explicit)

How did the young wife feel to bring up her baby? (Implicit)

**EQG**

Why did the wife go out to feed her pigs? (Explicit)

Why did the wife go out to feed her pigs? (Explicit)

How did the wife feel when she saw the pig? (Implicit)

How did the wife feel when she saw the pig? (Implicit)

**QAG**

how did the wife earn money? (Explicit)

what day was it? (Implicit)

what did the wife feed her pigs? (No Answer)

how many pigs did the wife plan to buy? (No Answer)

Table 15: Another example of generated questions of mQG, EQG, and QAG for a given context. On the right of each question, classified results by the answerability evaluation model are shown in parentheses. mQG model has generated questions most answerable questions with diversity.

You will be given a set of three questions. Please rank each set of questions. If it's hard to determine the superiority and inferiority, it's okay to give the same ranking, but please evaluate it as 1st to 3rd place as much as possible. (1: best, 3: worst)

A
Where did the prisoner go after he left the king's presence?
What did the king tell the young man to do after he left?
How did the young man feel after he left the king's presence?

B
who was the old woman?
what did the young man raise?
how long did the king give to bring back his son?

C
Where did the old woman appear near?
Where did the old woman appear near?
Where did the old woman appear near?

1. Which set of questions has more diverse question types? *

|   | 1 | 2 | 3 |
|---|---|---|---|
| A | ○ | ○ | ○ |
| B | ○ | ○ | ○ |
| C | ○ | ○ | ○ |

2. Which set of questions consists of more diverse phrases? *

|   | 1 | 2 | 3 |
|---|---|---|---|
| A | ○ | ○ | ○ |
| B | ○ | ○ | ○ |
| C | ○ | ○ | ○ |

3. Which set of questions can provide more diverse answers?                          *
(Give a better ranking for a set of questions that may require more complex reasoning to answer.)

|   | 1 | 2 | 3 |
|---|---|---|---|
| A | ○ | ○ | ○ |
| B | ○ | ○ | ○ |
| C | ○ | ○ | ○ |

Figure 6: The question sheet for diversity human evaluation.

Below is a paragraph from a story book. Four questions related to the story book are given and please rate each question. It's okay to give the same score. (1: worst, 5: best)

Then the poor man's heart grew less heavy, and he gave over worrying. So one fine day his rich neighbor came along with no fewer than twenty farmhands, and they mowed down one swath after another. But the poor neighbor did not even take the trouble to begin when he saw how the others took hold, and that he himself would not be able to do anything alone.

Questions
A: What will happen when the rich neighbor comes along with no fewer than twenty farmhands?
B: why did the poor neighbor not even take the trouble to begin mowing?
C: how long did it take for the rich neighbor to come along?
D: How did the poor neighbor feel when he saw how the others took hold?

1. The more grammatically correct questions, the higher the score. *

|   | 1 | 2 | 3 | 4 | 5 |
|---|---|---|---|---|---|
| A | ○ | ○ | ○ | ○ | ○ |
| B | ○ | ○ | ○ | ○ | ○ |
| C | ○ | ○ | ○ | ○ | ○ |
| D | ○ | ○ | ○ | ○ | ○ |

2. Determine if each question is answerable from the paragraph and give a higher *
score as you think it is possible.
a) If it's a question about what's going to happen in the future and you think it's possible to infer from the paragraph, please say it's possible to answer it.
b) Please rate the questions regardless of the difficulty level.

|   | 1 | 2 | 3 | 4 | 5 |
|---|---|---|---|---|---|
| A | ○ | ○ | ○ | ○ | ○ |
| B | ○ | ○ | ○ | ○ | ○ |
| C | ○ | ○ | ○ | ○ | ○ |
| D | ○ | ○ | ○ | ○ | ○ |

Figure 7: The question sheet for quality human evaluation.