# OpenReview forum: "Diversity Enhanced Narrative Question Generation for Storybooks"
_EMNLP/2023/Conference — EMNLP 2023 Main_

### Official Review · Reviewer_t52J · 2023-08-02

**Soundness:** 5

**Excitement:**

4: Strong: This paper deepens the understanding of some phenomenon or lowers the barriers to an existing research direction.

**Missing References:**

In section 2.2 (Diversity), what exactly are the few studies that addressed diversity as a challenge in QG?

**Paper Topic And Main Contributions:**

The main contributions of the paper, as perceived by me, are as follows:

- The introduction of a multi-question generation model that aims to generate diverse questions;
- An in-depth experimental study on the feasibility of the diversity question generator through an evaluation that includes several baselines, an answerability model, human feedback and zero-shot scenario;
- Ablation studies for further evaluating the generator robustness, allowing the reader to analyze the importance and interplay of different factors within the model.

**Questions For The Authors:**

I would like to put the following questions to the authors:

- (A) What are your thoughts on question diversity if adjusting the relative weights of LCE and LMQS in the overall loss function?
- (B) Still, related to the first question, is it feasible to modify the proposed training objective to incorporate the desired degree of diversity?
- (C) Have other diversity metrics been considered for evaluating question diversity? For example, see work by Li et al. (2016), which has been applied in Wang et al. (2022) and in Bulathwela et al. (2023).
- (D) Is there an idea on how to make the evaluation model improve the classification of implicit questions?
- (E) Following this research, what is the future work?

References:

Jiwei Li, Michel Galley, Chris Brockett, Jianfeng Gao, and Bill Dolan. 2016. A Diversity-Promoting Objective Function for Neural Conversation Models. In Proceedings of the 2016 Conference of the North American Chapter of the Association for Computational Linguistics: Human Language Technologies, pages 110–119, San Diego, California. Association for Computational Linguistics.

Wang, Z., Valdez, J., Basu Mallick, D., Baraniuk, R.G. (2022). Towards Human-Like Educational Question Generation with Large Language Models. In: Rodrigo, M.M., Matsuda, N., Cristea, A.I., Dimitrova, V. (eds) Artificial Intelligence in Education. AIED 2022. Lecture Notes in Computer Science, vol 13355. Springer, Cham. https://doi.org/10.1007/978-3-031-11644-5_13

Bulathwela, S., Muse, H., Yilmaz, E. (2023). Scalable Educational Question Generation with Pre-trained Language Models. In: Wang, N., Rebolledo-Mendez, G., Matsuda, N., Santos, O.C., Dimitrova, V. (eds) Artificial Intelligence in Education. AIED 2023. Lecture Notes in Computer Science(), vol 13916. Springer, Cham. https://doi.org/10.1007/978-3-031-36272-9_27

**Reasons To Accept:**

The strengths of the paper, as perceived by me, are as follows:

- The generation method. In particular, introducing a maximum question similarity loss (LMQS) is grounded and relevant for generating diverse questions, which is a challenge in question generation. The use of a recursive generation framework should also be highlighted positively;
- Robustness of evaluation. The evaluation is comprehensive, varied, and in-depth, with multiple studies attesting to the quality and diversity of the questions generated. In particular, the answerability model, and human evaluation should be highlighted positively. Zero-shot evaluation shows the applicability of the model in other contexts;
- The comparison and discussion of the results with recent baselines highlight the value of the work;
- The ablation studies and further analysis of the evaluation model are helpful in order to better understand the proposed model's limitations and applications.

**Reasons To Reject:**

The weaknesses of the paper, as perceived by me, are as follows:

- Lack of justification in the choice of the base model (BART) for building the final model (mQG). While BART has demonstrated remarkable performance in various NLP tasks, it has also been shown that, for example, T5 may be better for the specific task of question generation (Ushio et al., 2022). If BART is indeed the best choice, I suggest that the authors better substantiate its use;
- Lack of justification for combining LCE (cross-entropy loss) and LMQS (maximum question similarity loss). I wonder if the overall training objective may lead to a trade-off between promoting closeness to reference questions and promoting diversity in question generation. I suggest that the authors elaborate on this topic beyond just referring that cross-entropy loss is also used (at the end of Section 3.2);
- Measuring the quality of generated questions with Rouge-L F1 score. Specifically, Callison-Burch et al. (2006) and Liu et al. (2016) have shown that automatic metrics may not correlate well with quality since they essentially compute the n-gram similarity between the ground-truth and generated text. I suggest that the authors address the metric's limitations, particularly in this paper's context, where the aim is to generate diverse questions.

References:

Asahi Ushio, Fernando Alva-Manchego, and Jose Camacho-Collados. 2022. Generative Language Models for Paragraph-Level Question Generation. In Proceedings of the 2022 Conference on Empirical Methods in Natural Language Processing, pages 670–688, Abu Dhabi, United Arab Emirates. Association for Computational Linguistics.

Chris Callison-Burch, Miles Osborne, and Philipp Koehn. Re-evaluating the role of Bleu in machine translation research. In 11th Conference of the European Chapter of the Association for Computational Linguistics, pages 249–256, Trento, Italy, April 2006.

Chia-Wei Liu, Ryan Lowe, Iulian Serban, Mike Noseworthy, Laurent Charlin, and Joelle Pineau. How NOT to evaluate your dialogue system: An empirical study of unsupervised evaluation metrics for dialogue response generation. In Proceedings of the 2016 Conference on Empirical Methods in Natural Language Processing, pages 2122–2132, Austin, Texas, November 2016.

**Reproducibility:**

4: Could mostly reproduce the results, but there may be some variation because of sample variance or minor variations in their interpretation of the protocol or method.

**Reviewer Confidence:**

4: Quite sure. I tried to check the important points carefully. It's unlikely, though conceivable, that I missed something that should affect my ratings.

**Typos Grammar Style And Presentation Improvements:**

I suggest that the authors objectively state the contributions and findings in the Introduction.
The outline of the article can also be explained there.

In Figure 1, the arrows have three different colors, which come from boxes with other colors. I suggest that the authors use fewer colors for ease of following.

In Table 1 and others, the ranges for each metric are missing.

There are sentences where the word's first letter appears in the upper case, after a comma. For example, see page 1, line 78, line 8, line 571.

---

> ### Author Rebuttal · Authors · 2023-08-29
>
> Thank you for your thoughtful reviews and constructive suggestions!
>
> ### 1. Justification for choosing BART
> |     mQG    | # generated Questions Per Section | # Answerable Questions Per Section&uarr; | Rouge-L F1&uarr; | Self-BLEU&darr; |
> |:----------:|:---------------------------------------:|:----------------------------------------------:|:-------------------:|:---------------:|
> | T5-large  |                                   28.00 |                                          17.89 |               30.59 |      **0.5476** |
> | BART-large |                                   28.00 |                                      **23.35** |           **58.24** |          0.6243 |
>
> We appreciate your thoughtful consideration of choosing a base model for building the final model. To address this concern, we conducted an in-depth performance comparison between BART and T5 on the FairyTaleQA dataset. For T5, we have adopted the prefix (‘generate question: ’) by Ushio et al. (2022). As shown in the table, T5 generated a significantly low number of answerable questions and Rouge-L F1 score. We think that as T5 is already trained with multiple supervised tasks in a text-to-text manner, it is not fit for our specific way of training mQG. This substantiates our decision to utilize BART as the base model for building our final model, mQG.
> We will include these comparative results and the rationale for our choice in the camera-ready version if accepted.
>
> [1] Asahi Ushio, Fernando Alva-Manchego, and Jose Camacho-Collados. 2022. Generative Language Models for Paragraph-Level Question Generation. In Proceedings of the 2022 Conference on Empirical Methods in Natural Language Processing, pages 670–688, Abu Dhabi, United Arab Emirates. Association for Computational Linguistics.
>
> ### 2. Explanation on CE loss
> Thank you for the advice. In our observation, CE loss did play a vital role in making questions syntactically correct as it is calculated within the token level and mQS loss is calculated at the sentence level. Sorry for missing this information. We will add the essential role of CE loss in the camera-ready version if accepted.
>
> ### 3. Limitations of Rouge-L for Quality Evaluation
> Thank you for the suggestion to point out the limitation of the Rouge-L score. We looked through the reference papers you recommended. We totally agree that the Rouge-L score over ground-truth questions may not correlate to question quality. That is also the reason why we conducted a human evaluation on Table 3 to assess the quality of questions more comprehensively. In our human evaluation results regarding quality, there is a difference from the Rouge-L scores. All models have generated semantically correct questions. We will elaborate on this aspect and its connection to the limitations of Rouge-L score in the camera-ready version if accepted.
>
> |     | FairyTaleQA | TellMeWhy  | SQuAD      |
> |-----|-------------|------------|------------|
> | EQG | 0.9167      | 0.9129     | 0.8977     |
> | QAG | 0.9340      | 0.9342     | 0.9203     |
> | mQG | **0.9391**  | **0.9361** | **0.9210** |
>
> #### BERTScore (RoBERTa-large) Results
>
> Additionally, we have evaluated mQG on BERTScore [2] with Reviewers (Reviewer V2An and t52J) concern over Rouge-L score. On all datasets, mQG outperformed our main baselines. We will add these evaluation results on BERTScore in the camera-ready version if accepted.
>
> [2] Tianyi Zhang, Varsha Kishore, Felix Wu, Kilian Q. Weinberger, Yoav Artzi:
> BERTScore: Evaluating Text Generation with BERT. CoRR abs/1904.09675 (2019)
>
> ### Questions for the Authors
> We appreciate your insightful questions and suggestions for our future work. Here are our responses:
> ### A, B) Relative Weights Outcome
> Adjusting the relative weights to achieve higher diversity seems difficult at this time. The MQS loss operates at the sentence level, comparing reference questions. This setup might inherently limit the diversity to the diversity range of the reference questions.
>
> ### C) Diversity Metrics
> We sincerely appreciate your thoughtful suggestion to explore the distinct-n score as an additional diversity metric. While we recognize the merits of this metric and its potential to enhance our evaluation, we have chosen to focus solely on the self-BLEU metric for assessing question diversity. As we know, a distinct-n score counts distinct n-grams in a single sentence. Our task is to check diversity between questions so we think Self-BLEU is more favorable for our paper. We understand the potential benefits of considering additional diversity metrics and will certainly keep this suggestion in mind.
>
> ### D) Improving Implicit Question Classification
> + Shown in Table 7 (page 12), only about 27% of questions were implicit questions. We think the low accuracy of implicit questions compared to explicit questions is due to this low number of implicit questions.
> + Some conditions of unanswerable questions in SQuAD2.0 can be directed to the definition of implicit questions. This may cause model confusion.
> Due to the reasons mentioned above, we believe that the lower classification results observed for implicit questions can be attributed to issues within the datasets. However, problems stemming from dataset quality have been widely acknowledged within the realm of Natural Language Generation (NLG). We consider an approach that utilizes the Answer Generation (AG) model to address this challenge.
>
> The proposed strategy involves training the AG model to generate responses of "No Answer" for questions that are inherently unanswerable and to produce actual answers for questions that can be answered, regardless of whether they are explicit or implicit in nature. During the inference phase, a specific threshold probability for generating the "No Answer" response could be established. This threshold could serve as a criterion for identifying questions that are likely unanswerable. By adopting this methodology, we anticipate a potential enhancement in the categorization of questions, thus streamlining the process of obtaining accurate answers for such inquiries.
>
> ### E) Future Work
> Building on this research, we have identified potential directions for future work:
> + In this paper, we used explicit modeling of question types. Recognizing the complexity of language and the multitude of ways questions can be formulated, we acknowledge that our current modeling might not encompass all subtleties. To illustrate this point, consider the work by Olney et al. (2012), which delved into categorizing questions with a psychological aspect. We plan to apply this nuanced approach to question categorization, which we believe will further enhance the performance of mQG.
> + With the rapid and continuous advancements in LLMs, we are excited about the potential applications of our method. By extending our approach to exploit the power of LLMs, we can address a wide range of NLG tasks (summarization, dialogue, etc.) that demand outputs with varying degrees of complexity and diversity.
> + With our current answerability evaluation model, we encountered limitations in its ability to reliably classify implicit questions, which are inherently more challenging due to their nuanced nature. As implicit questions often lack explicit indicators, they can lead to misclassifications, affecting the overall performance. We will work on the AG model to make better classifications of questions into answerable or not.
>
> [3] Olney, Andrew M., Arthur C. Graesser and Natalie K. Person. “Question Generation from Concept Maps.” Dialogue Discourse 3 (2012): 75-99.
>
> ### Missing reference
> Sorry for the confusing reference. The few studies indicate three reference papers in lines 173~175. We will revise this missing part and make changes in the camera-ready version if accepted.
>
> ### Typos, Grammar, Style, and Presentation
> Thank you for pointing out areas where improvements are needed in our writing, figure design, and table formatting. We will make the following changes in the revised version.
> + Add an outline of the paper and objectively state contribution and findings in the introduction
> + Change the purple line into the blue line in Figure 1
> + Range explanation on each evaluation metrics
> + Check for additional typos

---

### Official Review · Reviewer_V2An · 2023-08-04

**Soundness:** 2

**Excitement:**

3: Ambivalent: It has merits (e.g., it reports state-of-the-art results, the idea is nice), but there are key weaknesses (e.g., it describes incremental work), and it can significantly benefit from another round of revision. However, I won't object to accepting it if my co-reviewers champion it.

**Missing References:**

[1] Diversify Question Generation with Continuous Content Selectors and Question Type Modeling https://aclanthology.org/2020.findings-emnlp.194/

**Paper Topic And Main Contributions:**

This paper introduces a multi-question generation method (mQG) that focuses on generating diverse and answerable questions from a given context. The proposed method uses BART to recursively generate questions based on the input of question type, context, and previously generated questions. It enforces a maximum question similarity loss, which is intended to promote similar hidden representations for different questions.

**Questions For The Authors:**

A. Are EQG and QAG based on BART-large as well?

**Reasons To Accept:**

- The proposed method increased the diversity of the generated questions for both in-domain and cross-domain settings, as measured by Self-BLEU.
- As suggested by human evaluation, the proposed method can indeed diversify the types of generated questions by specifying QT in the inputs.

**Reasons To Reject:**

- The authors claim that "These decoding methods tried to sample tokens from less likely vocabularies. This solution is less relevant in more constrained tasks such as question generation and machine translation, where beam search is a preferred method." and exclude all sampling-based methods from the comparison. However, previous works, including one of the authors cited work by Sultan et al, have demonstrated the effectiveness of sampling-based methods for diversifying question generation.
- The authors only used one token-based automatic evaluation metric, Rouge-L, for the quality evaluation, which may neglect semantic similarity and is not conclusive enough especially when the task is to increase the diversity.
- The Rouge-L score is calculated as "for each ground-truth question, we find the highest Rouge-L F1 score on generated questions from the same context than average overall scores.", which may result in multiple groud-truth questions getting matched to few generated questions and leads to an incomprehensive evaluation of the quality of all the generated questions.
- For the human evaluation part, most aspects have Krippendorf's alphas less than two-thirds, which indicates poor reliability. Only the question type aspect has a moderate level of reliability.
- The effectiveness of explicitly modeling question types for diverse question generation has been studied in previous work[1].
- Although the paper includes an ablation study that evaluated the architecture w/o the MQS loss, it remains unclear how the MQS loss affects the training and generations. It could be better if the authors can add a weighting factor and show the performance of the models in terms of question quality and diversity when the MQS loss is incorporated into the overall loss at different weights (particularly higher weights).

**Reproducibility:**

4: Could mostly reproduce the results, but there may be some variation because of sample variance or minor variations in their interpretation of the protocol or method.

**Reviewer Confidence:**

4: Quite sure. I tried to check the important points carefully. It's unlikely, though conceivable, that I missed something that should affect my ratings.

---

> ### Author Rebuttal · Authors · 2023-08-29
>
> Thank you for your thoughtful reviews and constructive suggestions!
>
> ### 1. Decoding methods
> Thank you for your insightful feedback regarding decoding methods. Here are some reasons we only used beam search:
> + Sultan et al. (2020) claim that sampling is more beneficial for improving QA model performance through QG. In fact, Sultan et al. (2020) argue that beam search performed better in terms of QG performance itself.
> + Sampling-based methods include randomness in each inference, leading to inconsistent outputs. Considering that mQG undergoes multiple rounds of inference for a single context, we believe producing consistent outputs is crucial. Hence, we opted to utilize beam search, which enables us to achieve this desired consistency.
> + Our main baselines, EQG and QAG papers, also exclusively employed beam search. Due to the fair comparison, we decided to follow their approach.
>
> We apologize for omitting these details in the paper and will incorporate them in the camera-ready version if accepted.
>
> [1] Md Arafat Sultan, Shubham Chandel, Ramón Fernandez Astudillo, and Vittorio Castelli. 2020. On the Importance of Diversity in Question Generation for QA. In Proceedings of the 58th Annual Meeting of the Association for Computational Linguistics, pages 5651–5656, Online. Association for Computational Linguistics.
>
>
> ### 2. Only used one token-based automatic evaluation metric, Rouge-L
> |     | FairyTaleQA | TellMeWhy  | SQuAD      |
> |-----|-------------|------------|------------|
> | EQG | 0.9167      | 0.9129     | 0.8977     |
> | QAG | 0.9340      | 0.9342     | 0.9203     |
> | mQG | **0.9391**  | **0.9361** | **0.9210** |
> #### BERTScore (RoBERTa-large) Results
>
> Thank you for bringing up this important point. We took your feedback and evaluated mQG on BERTScore [2]. This addition will allow us to consider semantic aspects that might not be adequately captured by Rouge-L.
>
> We have constructed the table with our two main baselines. We are pleased to share that even with BERTScore [2], mQG consistently outperformed the baseline methods in all datasets.  Your feedback has been helpful in refining our evaluation strategy to ensure a more robust and meaningful assessment of our approach. We will add this result in the camera-ready version if accepted.
>
> [2] Tianyi Zhang, Varsha Kishore, Felix Wu, Kilian Q. Weinberger, Yoav Artzi:
> BERTScore: Evaluating Text Generation with BERT. CoRR abs/1904.09675 (2019)
>
> ### 3. Calculation of Rouge-L score
> Appreciating the reviewer's insights, regarding the Rouge-L F1 score computed on the FairytaleQA test dataset using a combination set of ground-truth questions sharing context, the score is 29.06. This particular score highlights a scenario where minimal overlap exists among the ground-truth questions. Considering this result, if multiple ground-truth questions had indeed matched some generated questions, it's reasonable to assume that the resulting score would have been significantly lower across the performance of all models. We will add this information in the camera-ready version if accepted.
>
> ### 4. Reliability of human evaluation
> | Human Eval | Fleiss's kappa |
> |------------|---------------|
> | Type       |          0.53 |
> | Syntax     |          0.37 |
> | Content    |          0.26 |
> | Appro.     |          0.11 |
> | Ans.       |          0.15 |
>
> We appreciate the reviewer's observation regarding Krippendorff's alpha values in the human evaluation section.
> It's important to acknowledge that the reliability of Krippendorff's alpha can vary depending on the specific task being evaluated, the number of annotators, and the number of classes. Furthermore, it's also worth noting that even in the human evaluation by Cao and Wang (2021), which we followed, krippendorff's alpha did not exceed two-thirds. We have additionally calculated Fleiss’s kappa which is a similar inter-rater reliability test. Richard et al (1977) describe that Fleiss’s kappa score range between 0.20~0.40 indicates fair agreement on a condition of “two annotators, on two classes”. In the table, you can see all of our diversity evaluation results (Type, Syntax, Content) are above fair agreement even though we had five annotators to annotate three classes. For Appro. and Ans., as we had five annotators to rate on a five-point scale we believe it is an acceptable score.
>
> [3] Shuyang Cao and Lu Wang. 2021. Controllable Open-ended Question Generation with A New Question Type Ontology. In Proceedings of the 59th Annual Meeting of the Association for Computational Linguistics and the 11th International Joint Conference on Natural Language Processing (Volume 1: Long Papers), pages 6424–6439, Online. Association for Computational Linguistics.
>
> [4] Landis, J. Richard, and Gary G. Koch. “The Measurement of Observer Agreement for Categorical Data.” Biometrics, vol. 33, no. 1, 1977, pp. 159–74. JSTOR, https://doi.org/10.2307/2529310.
>
> ### 5. Weighting factor for MQS loss
> We expect that adjusting weights might not lead to a significant increase in diversity for a couple of reasons:
>
> + The MQS loss seeks the diversity of questions at the sentence level. This setup might inherently limit the diversity to the diversity range of the reference questions.
>
> + The MQS loss already contributes one-third of the total initial loss. Given that BART's architecture generates text token by token, token-level loss (Cross-Entropy loss) is essential for ensuring the syntactic correctness of the generated questions. We think the amount of MQS loss is high enough to get to the optimal points.
>
> ### Q. Are EQG and QAG based on BART-large as well?
> Following the implementation details described in their papers, we have used BART-large as their base model.
>
> ### Missing reference
> Thank you for pointing out the missing reference, which also explicitly modeled question types to generate questions.  Although Wang et al (2020)'s work is similar to mQG in utilizing question types there were some difficulties in using it as our baseline. Wang et al (2020) tried to get a question type distribution from context and answer pairs and use it to generate questions with different question types. Their approach of using answers as input additionally creates the need to annotate the answers for each context. This was not suitable for the large number of possible questions we were looking for. We will add this information in the related work of the camera-ready version if accepted.
>
> [5] Zhen Wang, Siwei Rao, Jie Zhang, Zhen Qin, Guangjian Tian, and Jun Wang. 2020. Diversify Question Generation with Continuous Content Selectors and Question Type Modeling. In Findings of the Association for Computational Linguistics: EMNLP 2020, pages 2134–2143, Online. Association for Computational Linguistics.

---

### Official Review · Reviewer_h6Ka · 2023-08-06

**Soundness:** 4

**Excitement:**

5: Transformative: This paper is likely to change its subfield or computational linguistics broadly. It should be considered for a best paper award. This paper changes the current understanding of some phenomenon, shows a widely held practice to be erroneous in someway, enables a promising direction of research for a (broad or narrow) topic, or creates an exciting new technique.

**Paper Topic And Main Contributions:**

This paper presents a multi-question generation model designed to produce diverse and answerable questions grounded in context. Notably, the L_mqs proposed by the authors emphasizes generating a variety of questions. The authors further undertake extensive research, employing both human and automated evaluations, to showcase the model's efficacy. For the automated diversity assessment, the authors also adopt the self-bleu metric.






**Reasons To Accept:**

The approach is innovative, particularly with the introduction of L_mqs to enhance the diversity of question generation. The authors have carried out thorough research, encompassing both human and automated evaluations, to validate its efficacy.






**Reasons To Reject:**

The experiment relies on BART, which is somewhat dated, and the size of BART is small. It would be beneficial if the authors could incorporate more experiments using recent advancements in LLM.






**Reproducibility:**

5: Could easily reproduce the results.

**Reviewer Confidence:**

3: Pretty sure, but there's a chance I missed something. Although I have a good feel for this area in general, I did not carefully check the paper's details, e.g., the math, experimental design, or novelty.

---

> ### Author Rebuttal · Authors · 2023-08-29
>
> Thank you for your valuable reviews and constructive suggestion!
>
> ### Experiment reliance on BART
>
> While we acknowledge that BART may be perceived as dated and relatively smaller, our current paper intentionally centers around a specific task to provide a focused analysis. In addition, we only used BART for a fair comparison, given that the baseline papers, QAG and EQG, also employed BART. Still, we fully agree that recent advancements in LLMs are crucial. We are eager to take this to future work with various tasks (summarization, conversation, etc.) requiring diversity.

---

### Meta-Review · Area_Chair_KQjB · 2023-09-18

**Recommendation:** 4

**Metareview:**

Main contributions:
* This paper introduces a multi-question generation model aimed at producing a wide array of contextually grounded, answerable questions with enhanced diversity. Notably, the LMQS (Maximum Question Similarity Loss) proposed by the authors underscores the generation of a variety of questions. The authors conduct a comprehensive research study, incorporating both human and automated evaluations, to demonstrate the model's effectiveness. In the context of automated diversity assessment, the authors also employ the Self-BLEU metric.

Reasons to accept:
* The approach demonstrates innovation, particularly through the introduction of LMQS to amplify the diversity in question generation. The authors have conducted extensive research, encompassing both human and automated evaluations, to validate its effectiveness. The ablation studies and further analysis of the evaluation model are helpful in order to better understand the proposed model's limitations and applications.

Reasons to reject:
* The use of BART was criticized as outdated and T5 was suggested as a more modern choice. In the rebuttal, the authors noted that T5 did not perform as well and will add details to the paper.
* Combination of cross entropy and the new LMQS. The authors addressed this concern by explaining that the mix with cross entropy was essential for better performance and will explain this in the final version.
* The use of ROUGE-L was criticized. In the rebuttal, the authors have addressed this concern with the use of BertScore.
* Krippendorff's alpha was criticized for being below two-thirds, but this has been shown to be typical for NLP tasks.

---

### Decision · Program_Chairs · 2023-10-07

**Decision:**

Accept-Main

**Comment:**

Main contributions:
* This paper introduces a multi-question generation model aimed at producing a wide array of contextually grounded, answerable questions with enhanced diversity. Notably, the LMQS (Maximum Question Similarity Loss) proposed by the authors underscores the generation of a variety of questions. The authors conduct a comprehensive research study, incorporating both human and automated evaluations, to demonstrate the model's effectiveness. In the context of automated diversity assessment, the authors also employ the Self-BLEU metric.

Reasons to accept:
* The approach demonstrates innovation, particularly through the introduction of LMQS to amplify the diversity in question generation. The authors have conducted extensive research, encompassing both human and automated evaluations, to validate its effectiveness. The ablation studies and further analysis of the evaluation model are helpful in order to better understand the proposed model's limitations and applications.

Reasons to reject:
* The use of BART was criticized as outdated and T5 was suggested as a more modern choice. In the rebuttal, the authors noted that T5 did not perform as well and will add details to the paper.
* Combination of cross entropy and the new LMQS. The authors addressed this concern by explaining that the mix with cross entropy was essential for better performance and will explain this in the final version.
* The use of ROUGE-L was criticized. In the rebuttal, the authors have addressed this concern with the use of BertScore.
* Krippendorff's alpha was criticized for being below two-thirds, but this has been shown to be typical for NLP tasks.